# LANDSCAPE LEARNING
# FOR NEURAL NETWORK INVERSION

## ABSTRACT

Many machine learning methods operate by inverting a neural network at inference time, which has become a popular technique for solving inverse problems in computer vision, robotics, and graphics. However, these methods often involve gradient descent through a highly non-convex loss landscape, causing the optimization process to be unstable and slow. We introduce a method that learns a loss landscape where gradient descent is efficient, bringing massive improvement and acceleration to the inversion process. We demonstrate this advantage on a number of methods for both generative and discriminative tasks, including GAN inversion, adversarial defense, and 3D human pose reconstruction.

## 1 INTRODUCTION

Many inference problems in machine learning are formulated as inverting a forward model $F(x)$ by optimizing an objective over the input space $x$. This approach, which we term optimization-based inference (OBI), has traditionally been used to solve a range of inverse problems in vision, graphics, robotics, recommendation systems, and security (Hernandez et al., 2008; Lee & Kuo, 1993; Domke, 2012; Brakel et al., 2013; Stoyanov et al., 2011; Cremer et al., 2019). Recently, neural networks have emerged as the paramterization of choice for forward models (Loper et al., 2015; Pavlakos et al., 2019; Abdal et al., 2019; Menon et al., 2020; Yu et al., 2021; Wang et al., 2019; Chen et al., 2021a; Zhang et al., 2021), which can be pretrained on large collections of data, and inverted at testing time in order to solve inference queries.

Optimization-based inference (OBI) has a number of advantages over feed-forward or encoder-based inference (EBI). Since there is no encoder, OBI provides flexibility to adapt to new tasks, allowing one to define new constraints into the objective during inference. When observations are partially missing, OBI can adapt without additional training. Moreover, OBI naturally supports generating multiple and diverse hypotheses when there is uncertainty. Finally, OBI has intrinsic advantages for robustness, both adapting to new data distributions as well as defending against adversarial examples.

However, the key bottleneck for OBI in practice is the computational efficiency and the speed of inference. Feed-forward models are fast because they only require a single forward pass of a neural network, but OBI requires many (often hundreds) steps of optimization in order to obtain strong results for one example. Forward models in OBI are often trained with generative or discriminative tasks, but they are not trained for the purpose of performing gradient descent in the input space. Fig. 8 visualizes the loss landscape for uncurated examples. The loss landscape is not guaranteed to be an efficient path from the initialization to the solution, causing the instability and inefficiency.

In this paper, we propose a framework to accelerate and stabilize the inversion of forward neural networks. Instead of optimizing over the original input space, we will learn a new input space such that gradient descent converges quickly. Our approach uses an alternating algorithm in order to learn the mapping between these spaces. The first step collects optimization trajectories in the new space until convergence. The second step updates the mapping parameters to reduce the distance between the convergence point and each point along the trajectory. By repeating these steps, our approach will learn a mapping between the spaces that allows gradient descent over the input to converge in significantly fewer steps.

Empirical experiments and visualizations on both generative and discriminative models show that our method can significantly improve the convergence speed for optimization. We validate our

approach on a diverse set of computer vision tasks, including GAN inversion (Abdal et al., 2019), adversarial defense (Mao et al., 2021), and 3D human pose reconstruction (Pavlakos et al., 2019). Our experiments show that our method converges an order of magnitude faster without loss in absolute performance after convergence. As our approach does not require retraining of the forward model, it can be compatible to all existing OBI methods with a differentiable forward model and objective function.

The primary contribution of this paper is an efficient optimization-based inference framework. In Sec. 2, we survey the related literature to provide an overview of forward model inversion problem. In Sec. 3, we formally define OBI (3.1); our method to learn an efficient loss landscape for OBI (3.2); a training algorithm for better generalization and robustness (3.3) . In Sec. 4, we experimentally study and analyze the effectiveness of the mapping network for OBI. We will release all code and models.

## 2 RELATED WORK

The different approaches for inference with a neural network can be partitioned into either encoder-based inference, which is feedforward, or optimization-based inference, which is iterative. We briefly review these two approaches in the context of our work. In addition, we also review representative work in meta-learning and discuss the similarities and differences with our work.

### 2.1 ENCODER-BASED INFERENCE

Encoder-based inference trains a neural network $F$ to directly map from the output space to the input space. Auto-encoder based approach (Pidhorskyi et al., 2020) learns an encoder that map the input data to the latent space. Richardson et al. (2021); Tov et al. (2021); Wei et al. (2021); Perarnau et al. (2016) learn an encoder from the image to the latent space in GAN. Encoder based inference requires training the encoder on the anticipated distribution in advance, which is often less effective and can fail on unexpected samples (Dinh et al., 2021; Kang et al., 2021).

### 2.2 OPTIMIZATION-BASED INFERENCE

OBI methods perform inference by solving an optimization problem with gradient-based methods such as Stochastic Gradient Descent (SGD) (Robbins & Monro, 1951) and Projected Gradient Descent (PGD) (Madry et al., 2017). In these cases, the objective function specifies the inference task. Besides these methods which use a point estimate for the latent variable, one can estimate the posterior distribution of the latent variable through Bayesian optimization, such as SGLD (Welling & Teh, 2011).

Gradient based optimization methods have been used to infer the latent code of query samples in deep generative models like GANs (Goodfellow et al., 2014) via GAN inversion (Karras et al., 2020; Jahanian et al., 2019; Shen et al., 2020; Zhu et al., 2016; Abdal et al., 2019; 2020; Bau et al., 2019; Huh et al., 2020; Pan et al., 2021; Yang et al., 2019). Style transfer relies on gradient based optimization to change the style of the input images (Jing et al., 2019). It can also create adversarial attacks that fool the classifier (Croce & Hein, 2020; Carlini & Wagner, 2017; Mao et al., 2019; Szegedy et al., 2013). Recently, back-propagation-based optimization has shown to be effective in defending adversarial examples (Mao et al., 2021) and compressed sensing (Bora et al., 2017). Wu et al. (2019) uses Model-Agnostic Meta-Learning (MAML) (Finn et al., 2017) to accelerate the optimization process in compressed sensing.

Recently, constrained optimization was popularized for text-to-image synthesis by Crowson et al. (2022); Liu et al. (2021). They search in the latent space to produce an image that has the highest similarity with the given text as measured by a multi-modal similarity model like CLIP (Radford et al., 2021). Test-time constrained optimization is also related to the idea of 'prompt-tuning' for large language models. Lester et al. (2021) learns "soft prompts" to condition frozen language models to perform specific downstream tasks. Soft prompts are learned through backpropagation to incorporate signal from just a few labeled examples (few-shot learning).

A major challenge for optimization-based inference is how to perform efficient optimization in a highly non-convex space Geiger et al. (2021). To address this, input convex model (Amos et al., 2017) was proposed so that gradient descent can be performed in a convex space. Tripp et al. (2020) introduced a method to retrain the generative model such as it learns a latent manifold that is easy

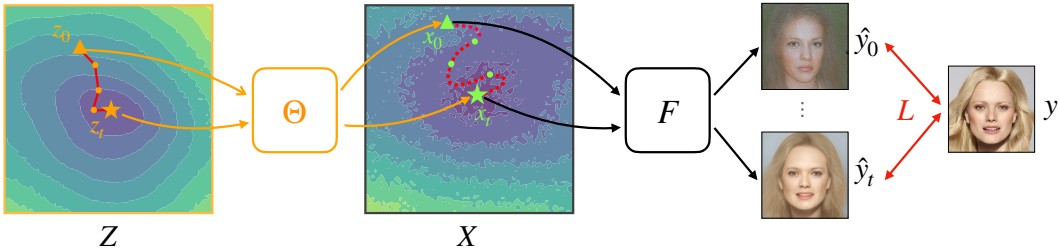

Figure 1: **Method**. The left and middle figure show the loss landscape for our latent space and the original latent space, respectively. While walking to the optimal solution in a few steps is hard in $X$ space, it can be done in our learned loss landscapes.

to optimize. When the model cannot be changed and retrained, bidirectional inference (Wang et al., 2019) and hybrid inference (Zhu et al., 2016; 2020) uses an encoder to provide a good initialization for the optimization-based inference in a non-convex space. Soori et al. (2021) proposes a time-efficient Natural GRadient Descent method for solving inverse of neural network's Fisher information matrix. Our method does not retrain the generative model, but instead maps a new latent space into the original input space, allowing more efficient optimization.

### 2.3 META-LEARNING

Given a distribution of tasks, meta-learning aims to adapt quickly when presented with a previously unseen task Hospedales et al. (2021). MAML (Finn et al., 2017) and related methods (Finn et al., 2018; 2019) propose a method to learn a parameter initialization of a neural network by differentiaing through the fine-tuning process. To reduce the computational cost of MAML due to the 2nd degree gradient, Nichol et al. (2018) proposes a first-order meta-learning algorithm. Unlike MAML which include modifying the forward model (e.g. in Wu et al. (2019)), our approach is able to keep the forward model fixed in order to maintain its learned rich priors. Instead of meta-learning of model initialization, a line of work (Andrychowicz et al., 2016; Ravi & Larochelle, 2016; Chen et al., 2021b; Wichrowska et al., 2017) proposed to learn an optimizer, often in the form of an RNN, in order to accelerate gradient descent. Unlike learned optimizers that try to create better optimization algorithms, our approach instead learns to remap the loss landscape which is compatible with any choice of optimizer, including standard SGD or learned ones.

### 3 LEARNING LANDSCAPES FOR EFFICIENT INFERENCE

We present our framework to learn an efficient loss landscape for optimization-based inference (OBI) methods. In Sec. 3.1, we will define OBI. In Sec. 3.2, we will introduce our framework as well as the training objective. In Sec. 3.3, we will describe how to train our model with an alternating optimization algorithm and an experience-replay buffer.

### 3.1 OPTIMIZATION-BASED INFERENCE

Let $F(\mathbf{x}) = \hat{y}$ be a differentiable forward model that generates an output $\hat{y}$ given an input variable $\mathbf{x} \in X$. For example, $\hat{y}$ might be an image, and $\mathbf{x}$ might be the latent variables for a generative model. Given an observation $y$, the goal of OBI is to find the input $\hat{\mathbf{x}} \in X$ such that an objective function $L(\hat{y}, y)$ is minimized. Formally, we write this procedure as:

$$\hat{\mathbf{x}} = \operatorname*{argmin}_{\mathbf{x} \in X} L(F(\mathbf{x}), y) \tag{1}$$

When the objective function $L$ and the model $F$ are both differentiable, we can perform the optimization over input space $X$ with gradient descent.

### 3.2 REMAPPING THE INPUT SPACE

Instead of operating in the original input space $X$, we will create a new space $Z$ where gradient descent is efficient and converges in a small number of iterations. We will use a neural network

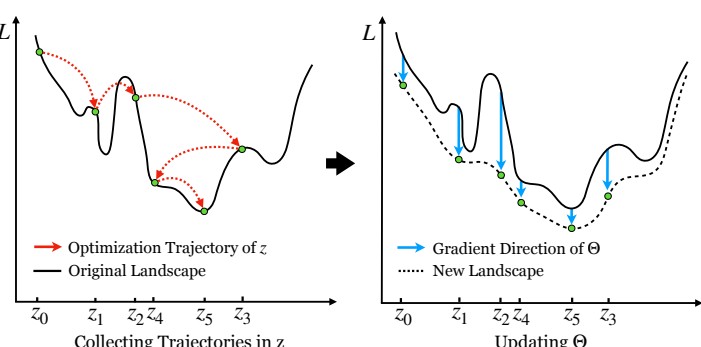

Figure 2: **Landscape Learning.** An optimization trajectory $\{\mathbf{z}_t\}_{t=0}^5$ collected is used to train $\Theta$. $\mathbf{z}_i$ that corresponds to a higher $L_i$ will yield a higher gradient when training $\Theta$. Optimization over multiple steps along the trajectory causes $\Theta$ to learn *patterns* of trajectories and create a more efficient loss landscape.

$\Theta : Z \to X$ that maps from the new space $Z$ to the original space $X$. The parameters of the mapping $\Theta$ is the vector $\theta$. The learning problem we need solve is to estimate $\theta$ so that there is a short gradient descent path in $Z$ from the initialization to the solution. Fig. 1 shows an overview of this setup.

We formulate an objective by rolling out the gradient updates on $\mathbf{z}$, where we write $\mathbf{z}_t \leftarrow \mathbf{z}_{t-1} + \lambda \frac{\partial L}{\partial \mathbf{z}_{t-1}}$ as the $t^{\text{th}}$ update with a step size of $\lambda$. For gradient descent in space $Z$ to be efficient, the goal of our introduced $\Theta$ is to move every step $\mathbf{z}_t$ as close as possible to the global minima:

$$\hat{\theta} = \underset{\theta}{\arg\min} \ \mathbb{E}_{(\mathbf{z},y)} \left[ \sum_{t=0}^T L(F(\Theta(\mathbf{z}_t)), y) \right] \quad \text{where} \quad \mathbf{z}_t = \begin{cases} 0, & t = 0 \\ \mathbf{z}_{t-1} + \lambda \frac{\partial L}{\partial \mathbf{z}_{t-1}}, & t > 0 \end{cases} \quad (2)$$

We visualize this process with a toy example in Fig. 2. Gradient updates on $\Theta$ w.r.t multiple steps $\mathbf{z}_t$ along a trajectory will cause the loss value on each step to be lowered. By learning the parameters of $\Theta$ across many examples, $\Theta$ can learn the patterns of optimization trajectories in $X$. For example, $\Theta$ can learn to estimate the step size in $X$ and dynamically adjust it to each example. Moreover, $\Theta$ can learn to reduce non-convexity of the landscape.

Once we obtain $\hat{\Theta}$ with parameters $\hat{\theta}$, we do inference on a new example $y$ through the standard optimization-based inference procedure, except in $Z$ space now. Given the observation $y$, we find the corresponding $\hat{\mathbf{x}}$ through the optimization:

$$\hat{\mathbf{x}} = \hat{\Theta}(\hat{\mathbf{z}}) \quad \text{where} \quad \hat{\mathbf{z}} = \underset{\mathbf{z} \in Z}{\arg\min} \ L(F(\hat{\Theta}(\mathbf{z})), y) \quad (3)$$

When the inverse problem is under-constrained, one can infer multiple hypotheses for $\hat{\mathbf{x}}$ by repeating the above optimization multiple times with a different random initialization for $\mathbf{z}_0$.

### 3.3 TRAINING

We use alternating optimization (AO) in order to train $\Theta$ jointly with estimating $\mathbf{z}$ for each example in the training set. Specifically, we first fix parameters of $\Theta$ and collect $N$ optimization trajectories of $\mathbf{z}$, each with length $T$. Adopting the same terminology from the robotics community for learning on continuous states Mnih et al. (2013), we term this an *experience replay buffer*. Subsequently, we randomly sample data from this buffer and train $\Theta$ to optimize the loss function. We alternate between these two steps for a fixed number of iterations with gradient descent for both. Depending on the application, the training time for $\Theta$ varied from one hour to one day using a four GPU server. The complete algorithm is provided in algorithm 1. Please see the appendix for more implementation details.

We also experimented with an online version of the training algorithm, where we update parameters of $\Theta$ immediately after one update to $\mathbf{z}$. However, in our experiments, we found this resulted in a slower convergence rate. We show these comparisons in the ablation experiments.

## 4 EXPERIMENTS

The goal of our experiments is to analyze how well our proposed method can be applied to various existing OBI methods to improve the optimization efficiency. We demonstrate application of our method to three diverse OBI methods in computer vision, including both generative models and

---

**Algorithm 1** Learning Mapping Network $\Theta$

---

1: **Input:** Ground truth $y$, step size $\lambda_z$ and $\lambda_\theta$, number of buffers $B$, number of data samples in a buffer $N$, number of optimization steps per sample $T$, loss function $L$, and forward model $F$.
2: **Output:** Learned mapping network $\Theta$
3: Randomly initialize a mapping network $\Theta$ with parameters $\theta$
4: **for** $b = 1, ..., B$ **do**
5:     Initialize Experience Replay Buffer $[\{\mathbf{z}_{t,i}\}_{t=1}^{T}]_{i=1}^{N}$
6:     **for** $i = 1, ..., N$ **do**
7:         $\mathbf{z}_0 \leftarrow \mathbf{0}$
8:         **for** $t = 1, ..., T$ **do**          Collecting optimization trajectory in $Z$ while keeping $\theta$ fixed
9:             $\mathbf{z}_t \leftarrow \mathbf{z}_{t-1} + \lambda_z \frac{\partial}{\partial \mathbf{z}_{t-1}} L(F(\Theta(\mathbf{z})), y)$
10:             $\mathbf{z}_{t,i} \leftarrow z_t$
11:         **end for**
12:     **end for**
13:     **for** $j = 1, ..., T \cdot N$ **do**          Updating $\theta$ with previously collected latents
14:         Randomly sample $\mathbf{z}$ from previously collected buffer
15:         $\theta_j \leftarrow \theta_{j-1} + \lambda_\theta \frac{\partial}{\partial \theta_{j-1}} L(F(\Theta(\mathbf{z})), y)$
16:     **end for**
17: **end for**
18: Return $\theta = 0$

---

discriminative models. For each OBI method, the inference-time optimization objective of the baseline and ours can be written as:

$$\text{Baseline: } \hat{\mathbf{x}} = \min_{\mathbf{x} \in X} L(F(\mathbf{x}), y), \quad \text{Ours: } \hat{\mathbf{z}} = \min_{\mathbf{z} \in Z} L(F(\Theta(\mathbf{z})), y) \tag{4}$$

Next, we provide the specific implementation of the loss term $L$ for each application, along with quantitative and qualitative results. We also perform experiments to understand the loss landscape in Sec. A and perform ablations on different parts of our approach in Sec. 4.4.

## 4.1 GAN INVERSION

We first validate our method on StyleGAN inversion (Abdal et al., 2019). We take a pretrained generator of StyleGAN Karras et al. (2019) denoted as $F$. Let $y$ be an observed image whose input variable we are recovering, we optimize the objective of Eq. 4 where the loss can be written as,

$$L(\hat{y}, y) = L_{lpips}(\hat{y}, y) + ||\hat{y} - y||_2^2 \tag{5}$$

where $L_{lpips}$ is a perceptual similarity loss introduced in Zhang et al. (2018), $\hat{y} = F(\hat{x})$ for baseline and $\hat{y} = F(\Theta(\hat{z}))$ for ours. We train $\Theta$ on the train split of CelebA-HQ (Karras et al., 2017) dataset and evaluate on CelebA-HQ validation split for in-distribution experiments and LSUN-Cat(Yu et al., 2015) for distribution shifting (OOD) experiments. We compare the results from our method against the state-of-the-art encoder-based GAN inversion model (Richardson et al., 2021).

**Quantitative Results.** From Fig. 3a, we see that in all experiments, optimization in our space $Z$ consistently outperforms the baseline from the first optimization step to after convergence. This gap in performance is even larger when evaluated on OOD data. This suggests that the improvement in performance is not caused by memorizing the training data. Note that our image reconstruction performance after only 2 steps of optimization is able to outperform or be on-par with the baseline at 20 steps of optimization, resulting in a 10-fold improvement in efficiency. Even after convergence (after 2000 optimization steps), our reconstruction performance improves over the baseline by 15% for in-distribution evaluation and 10% for OOD evaluation. When compared with encoder-based GAN inversion (Richardson et al., 2021), our method achieves better reconstruction after 11 steps of optimization for in-distribution data and 3 steps for OOD data.

**Qualitative Results.** From Fig. 7, we can see that our method already shows improvements on in-distribution data - it can almost perfectly reconstruct details like fine hair strands, the cap on the person's head, the object in the person's mouth as well-as the text on it. Interestingly, our method is able to discover and reconstruct latents for cats while the encoder-based model fails miserably as

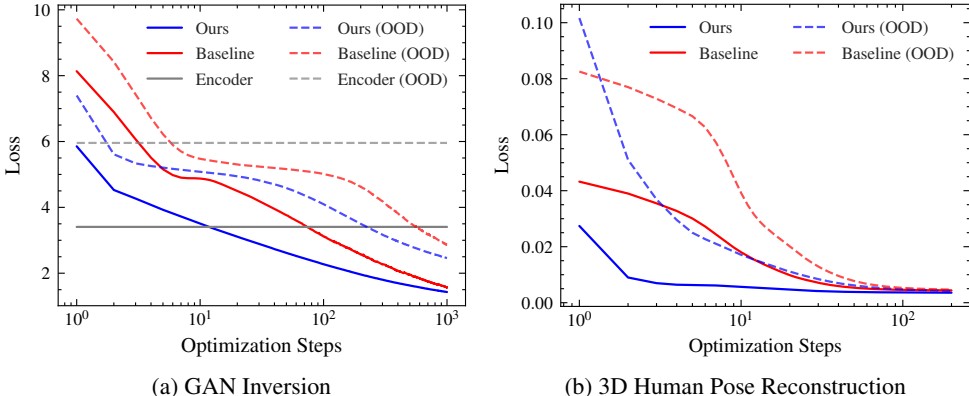

(a) GAN Inversion  (b) 3D Human Pose Reconstruction

Figure 3: **Optimization Performance.** We visualize the trends of optimization performance compared with the baseline. In **GAN Inversion (Left)**, we evaluate all models on test splits of CelebA-HQ (Karras et al., 2017) and LSUN-cat (Yu et al., 2015) (OOD) with loss defined in Eq. 5. Since encoder-based inference doesn't involve optimization, we use a flat line to represent it. We perform 2000 steps of gradient descent for all models except encoder-based models. In **3D Human Pose Reconstruction (Right)**, we evaluate all models on test splits of GRAB (Taheri et al., 2020) and PROX (Hassan et al., 2019) (OOD) with loss defined in Eq. 6. We perform 200 steps of gradient descent for all models. For each step, we plot the average loss value of test splits.

shown in Fig 7. The performance on OOD data truly highlights the benefits of our method. We also visualize how the face generations evolve over the process of optimization in Fig. 4. We can see that in just 4 steps, our method is already able to reconstruct coarse features such as face orientation, skin tone and hair color, while the baseline has hardly deviated from the initialization in regard to any of these features. Further, in Fig. 6 we visualize reconstructions from partial observations where only the center of the face (row 1) or everything other than the mouth (row 2) is visible. We can see a variety of feasible possibilities for the hidden regions (e.g., different hairstyles, lip colors, expressions, etc) showcasing the diversity of the new latent space. More uncurated examples of image inpainting and outpainting can be found in Fig. 9.

### 4.2 3D HUMAN POSE RECONSTRUCTION

In addition to image generation, our framework also works for 3D reconstruction. For this, we use VPoser (Pavlakos et al., 2019) – a variational autoencoder (Kingma & Welling, 2013) that has learnt a pose prior over body pose. VPoser was trained on SMPL-X body pose parameters $y \in \mathbb{R}^{63}$ obtained by applying MoSh (Loper et al., 2014) on three publicly available human motion capture datasets: CMU Mocap CMU, training set of Human3.6M (Ionescu et al., 2014), and the PosePrior dataset (Akhter & Black, 2015).

We take a pretrained VPoser decoder denoted as $F$. Our trained mapping network $\Theta$ projects a vector from the new input space $\mathbf{z} \in Z$ to a vector in the original VPoser decoder's input space $\mathbf{x} \in X$. Similar to GAN Inversion, we optimize the objective of Eq. 4, where the loss function between predicted and ground truth pose parameters is,

$$L(\hat{y}, y) = ||\hat{y} - y||_2^2 \tag{6}$$

where $\hat{y} = F(\hat{x})$ for the baseline and $\hat{y} = F(\Theta(\hat{z}))$ for ours. For training $\Theta$, we use the GRAB dataset (Taheri et al., 2020) which contains poses of humans interacting with everyday objects. We construct splits for novel video sequences – thus the test split will contain a seen human subject but a potentially unseen pose / demonstration by that subject. We evaluate on this test split for in-distribution experiments and on the PROX dataset (Hassan et al., 2019) for OOD experiments, which contains poses of humans interacting in 3D scenes (e.g., living room, office, etc).

**Quantitative Results.** For SMPL-X human pose reconstruction experiment, the results follow a similar trend as GAN inversion, with massive improvement in both convergence speed and final loss values after convergence (see Fig. 3b). Our method outperforms the baseline by 19% for in-distribution evaluation and 11% for OOD evaluation.

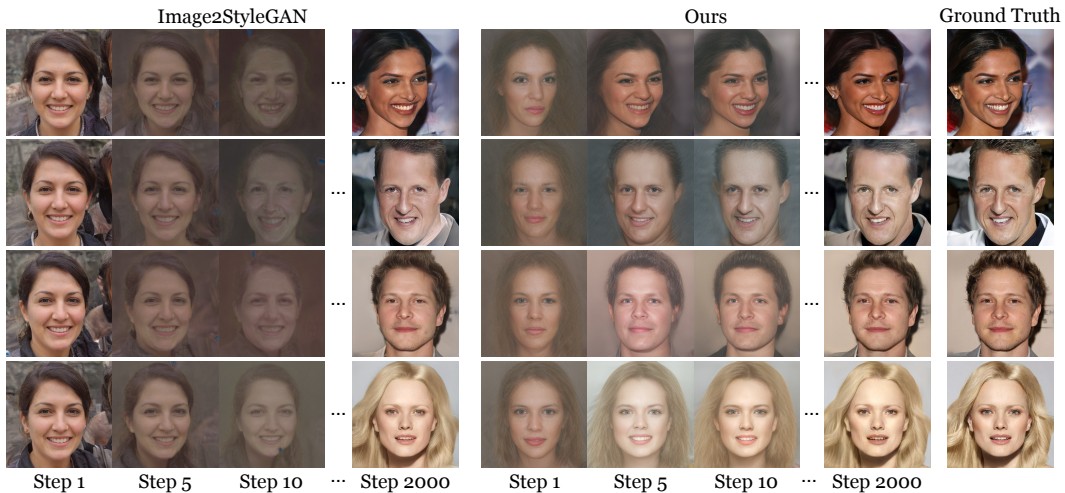

Figure 4: **Optimization Process for GAN Inversion.** Comparing optimization process of our method and the baseline in order to reconstruct the ground truth image. **Left** shows the results from the baseline where optimization is done in the original input space $X$. **Middle** shows the results from our method where optimization is done in our space $Z$. **Right** column contains the ground truth image to each example. Each row corresponds to the same example.

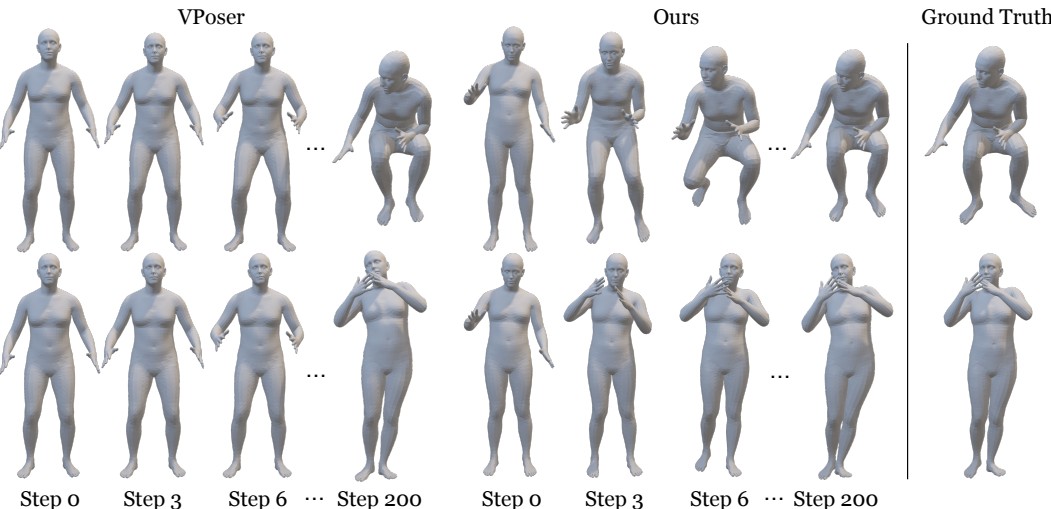

Figure 5: **Optimization Process for 3D Human Pose Reconstruction.** Results shown are for out-of-distribution PROX dataset for sitting (**Top**) and standing (**Bottom**) poses.

**Qualitative Results.** In Fig. 5 we visualize how the human pose reconstructions evolve over the process of optimization. Here, we observe that the reconstructions from steps 0 to 6 of the baseline are similar for both examples. On the other hand, our method caters to fast convergence for the varying examples, highlighting the general, yet efficient properties of our search space. Further, in Fig. 6 we visualize reconstructions from partial observations where the only joints visible are that of the upper body (row 3) or lower body (row 4). We obtain a wide range of feasible possibilities for the hidden joints demonstrating the diversity of the latent space.

## 4.3 DEFENDING ADVERSARIAL ATTACKS

Our method is also applicable to discriminative models. A state-of-the-art defense (Mao et al., 2021) for adversarial attack optimizes the self-supervision task at inference time, such that the model can dynamically restore the corrupted structure from the input image for robust inference. Following the existing algorithm implementation, we optimize the input image via our method by minimizing the

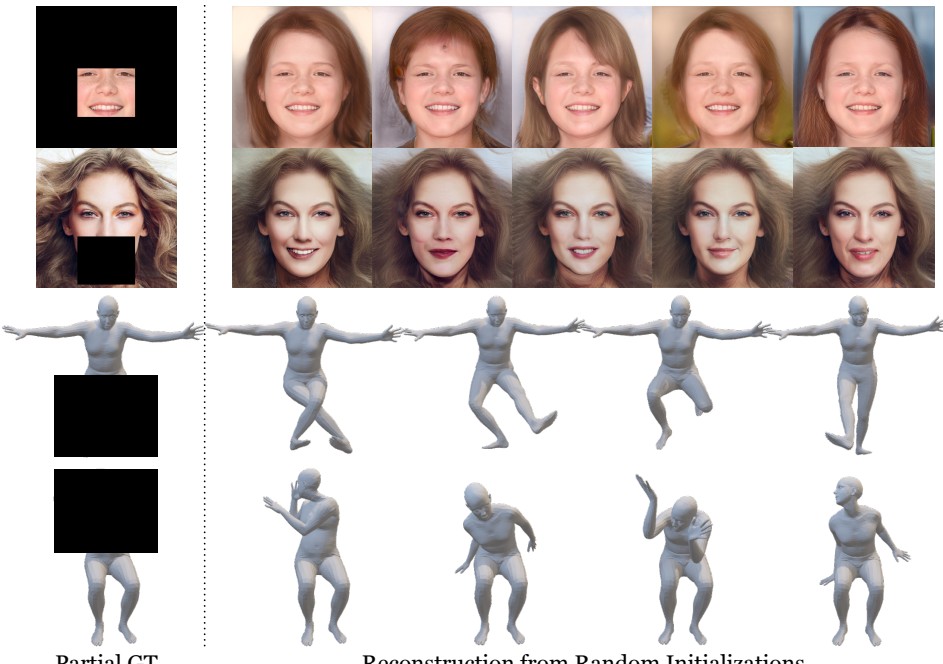

Partial GT          Reconstruction from Random Initializations

Figure 6: **Diversity of Masked Reconstructions.** We visualize reconstructions for partially observable inputs from random initializations. The masked regions are not considered for loss computation, i.e., the gradient is set to be zero. By optimizing only on the partial observation, we obtain diverse, feasible solutions for the hidden regions. 5 or 4 out of 20 most representative samples are presented.

following discriminative loss function:

$$L(F(\mathbf{r} + \mathbf{a}), y) = L(F(\Theta(\mathbf{z}) + \mathbf{a}), y) = \mathbb{E}_{i,j} \left[ -y_{ij}^{(s)} \log \frac{\exp(\cos(\mathbf{f}_i, \mathbf{f}_j))}{\sum_k \exp(\cos(\mathbf{f}_i, \mathbf{f}_k))} \right] + \lambda ||\Theta(\mathbf{z})||_2^2, \quad (7)$$

where $\mathbf{a}$ is the adversarial attacks that we aim to defend against, $\mathbf{r} = \Theta(\mathbf{z})$ is our additive defense vector to optimize, $\mathbf{f}_i$ are the contrastive features produced by the neural network $F$ from the $i^{\text{th}}$ image instance, and $\mathbf{y}_{ij}^{(s)}$ is the indicator for the positive pairs and negative pairs.

After obtaining the mapping network $\Theta$ and the input variable $\hat{\mathbf{z}}$, the prediction is $\hat{\mathbf{y}} = F'(\mathbf{a} + \Theta(\mathbf{z}))$, where $F'$ is the classification model. Note that the a self-supervision loss is optimized as a proxy for increasing the robust classification accuracy. In addition, we add a $L_2$ norm decay term for the generated noise $z$ to avoid generating reversal vector that is too large.

| Model Type | No Optimization | Optimization Steps | | | | | |
|---|---|---|---|---|---|---|---|
| | | 1 step | | 3 steps | | 5 steps | |
| | | Baseline | Ours | Baseline | Ours | Baseline | Ours |
| RO | 31.99 | 34.62 | **44.65** | 36.77 | **44.23** | 38.38 | **43.43** |
| AWP | 35.61 | 39.54 | **51.39** | 42.81 | **51.67** | 44.96 | **51.05** |
| MART | 35.66 | 39.77 | **51.77** | 42.50 | **51.77** | 45.42 | **50.96** |
| SemiSL | 29.78 | 34.53 | **52.11** | 37.27 | **51.23** | 40.93 | **49.83** |

Table 1: Experiment on improving adversarial robust accuracy. Our goal is to defend 200 steps of $L_2 = 256/255$ norm bounded attack (Madry et al., 2017), where the attack's step size is $64/255$. Our baseline is the SOTA test-time optimization-based defense (Mao et al., 2021), which minimizes the loss of self-supervision task.

**Quantitative Results.** We evaluate our method on four popular pretrained robust models (Rice et al., 2020; Wang et al., 2020; Wu et al., 2020; Carmon et al., 2019) on CIFAR-10 dataset. The results in Mao et al. (2021) require many steps to optimize the objective to improve the adversarial robustness, which slows down the inference by hundreds of times than the original forward pass. Ideally, we desire test-time optimization that can adapt to the attacked images in just one step, causing

In-Distribution        Out-of-Distribution

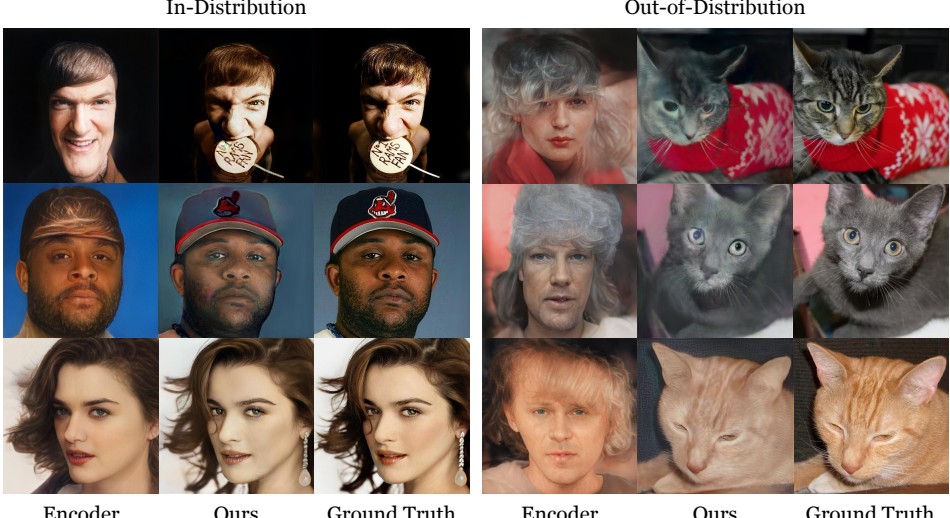

Encoder    Ours    Ground Truth    Encoder    Ours    Ground Truth

Figure 7: **Comparison Against Encoder-Based Inference. Left** shows the results on the test split of CelebA-HQ; **Right** shows the results on the LSUN-cat dataset.

| | Number of Steps | Full Model | Without AO and Buffer | Random Θ | Baseline |
|---|---|---|---|---|---|
| In-distribution | 20 | 3.082 | 3.583 | 4.417 | 4.456 |
| OOD | 20 | 4.932 | 5.127 | 5.135 | 5.292 |
| In-distribution | 200 | 1.964 | 3.034 | 2.723 | 2.569 |
| OOD | 200 | 3.498 | 4.756 | 3.823 | 4.617 |

Table 2: **Ablation Study on mapping network.** Number of Steps indicates the number of optimization steps performed during inference.

the minimal delay at inference time. In Table 1, our method outperforms the gradient descent method in Mao et al. (2021) by up to 18% robust accuracy using a single step, providing a real-time robust inference method. Note that our method converges after 1 step of optimization, demonstrating the effectiveness of our approach.

### 4.4 ABLATION STUDY

In this section, we present an ablation study by removing the proposed alternating optimization (AO) scheme and the experience replay buffer. We also compare against a Θ that is randomly initialized. From Table 2 in appendix, we discovered that for in-distribution, AO and training of Theta improves the optimziation performance by 14% and 30% respectively. Such gap becomes 35% and 28% for evaluation on 200 steps. For OOD data, the advantage is furthered enlarged as shown in Table 2.

One surprising result we discovered experimentally is that OBI under a randomly initialized mapping network Θ consistently outperforms the baseline. We believe this is due to the fact that adding a Gaussian distribution to an underlying latent distribution of StyleGAN is beneficial in smoothing out loss landscape, making it easier to perform OBI. Similar random projection can also be found in Wu et al. (2019).

### 5 CONCLUSION

This paper presents a method to accelerate optimization-based inference to invert a forward model. We propose an approach that learns a new space that is easier than the original input space to optimize with gradient descent at testing time. Our experiments and analysis on three different applications in computer vision have shown that by learning this mapping function, optimization becomes more efficient and generalizes better to out-of-distribution data. Through quantitative and qualitative analysis, we found that such improvement in optimization performance comes from a more efficient loss landscape. Since optimization-based inference has many advantages over encoder-based inference, we believe methods to accelerate them will have many impacts in a variety of applications.

## 6 ETHICS STATEMENT

Optimization-based inference has a wide variety of applications broadly in computer vision, robotics, natural language processing, assistive technology, security, and healthcare. Since our proposed method provides significant acceleration to these inference techniques, we expect our work to find positive impact in these applications, where speed, accuracy, and robustness are often critical.

Our algorithm is compatible with many different types of forward models – as long as a gradient can be calculated – including neural networks. However, learned forward models are known to acquire and contain biases from the original dataset. Our approach will also inherit the same biases. While our approach to inference offers some advantages to out-of-distribution data, it does not mitigate nor correct biases. The datasets in our experiments are likely not representative of the population, and consequently also contain biases. We acknowledge these limitations, and applications of this method should be mindful of these limitations, especially in potentially sensitive scenarios. As the community continues to address biases in models and datasets, our method can be applied to these models in the future too.

As authors of the paper, we have carefully reviewed and will adhere to the code of ethics provided at `https://iclr.cc/public/CodeOfEthics`.

## 7 REPRODUCIBILITY STATEMENT

We will release all code, models, and data. Here we provide the implementation details.

### 7.1 GAN INVERSION

Mapping network $\Theta$ is implemented with a 3-layer MLP. The input dimension (dimension of $Z$ space) is the same as the output dimension (dimension of $X$ space). For each intermediate output, we apply a Leaky ReLu function with a negative slope of 0.2 as an activation function. Hidden dimension of the MLP is 1024. For optimizing $z$ (collecting optimization trajectories), we use an Adam optimizer with a learning rate of 0.1. For training the mapping network $\Theta$, we use an AdamW optimizer with a weight decay of 0.1 with a learning rate of 0.0001. We used the following parameter set, $T = 20$, $N = 256$, $B = 500$. For baseline, we use the implementation of Abdal et al. (2019) provided here. The pretrained weights of StyleGAN converted to PyTorch are also provided in the same link.

### 7.2 3D HUMAN POSE RECONSTRUCTION

Mapping network $\Theta$ is implemented with a 3-layer MLP. The input dimension is 128 (dimension of $Z$) and the output dimension is 32 (dimension of $X$). For each intermediate output, we apply a Leaky ReLu function with a negative slope of 0.2 as an activation function. Hidden dimension of the MLP is 512. For optimizing $z$ (collecting optimization trajectories), we use an Adam optimizer with a learning rate of 0.1. For training the mapping network $\Theta$, we use an AdamW optimizer with a weight decay of 0.1 and a learning rate of 0.005. We use the following parameter set, $T = 200$, $N = 40960$, $B = 500$. For a fair comparison with the baseline, we tried a range of learning rate for $\mathbf{x} \in X$ $\{0.5, 0.1, 0.05, 0.01, 0.001\}$ and select the best performing configuration for comparison.

### 7.3 DEFENDING ADVERSARIAL ATTACKS

Mapping network $\Theta$ is implemented with a 3-layer MLP. The input dimension is 3072 (dimension of $Z$) and the output dimension is 3072 (dimension of $X$). For each intermediate output, we apply a Leaky ReLu function with a negative slope of 0.2 as an activation function. Hidden dimension of the MLP is 3072. For optimizing $z$ (collecting optimization trajectories), we use an Adam optimizer with learning rate $0.2/255$. For training the mapping network $\Theta$, we use an AdamW optimizer with learning rate of 0.0001 and a weight decay of 0.1. We use the following parameter set, $T = 5$, $N = 5120$, $B = 70$. For the regularization term $\lambda$ that constrains the amplitude of the additive defense vector, we use $\lambda = 1$. We use random start instead of zero start for initializing the attack reversal vector.

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

## APPENDIX

## A  LOSS LANDSCAPE VISUALIZATION

To understand the underlying cause of the significant improvement in optimization brought by our mapping network $\theta$, we visualize in Fig. 8 the loss landscape for performing optimization in the original input space $X$, and our projected space $Z$. To generate this visualization, we first perform 20 steps of optimization on the validation dataset to collect a set of recovered latents. We then perform principle component analysis (PCA) on these recovered latents to obtain two principle directions. Finally, for individual examples, we evaluate the loss for vertices on a meshgrid spanned by the two principle directions where the center is the last step of optimization.

From the visualization, we can see that the loss landscapes of the baseline are highly non-convex and contain points whose loss are significantly higher than its neighboring regions, while our loss landscapes are significantly smoother, with the "spikes" removed. Besides, our loss landscapes also tend to be steeper than the baseline ones. These two phenomena directly cause our method to perform gradient descent faster and stabler.

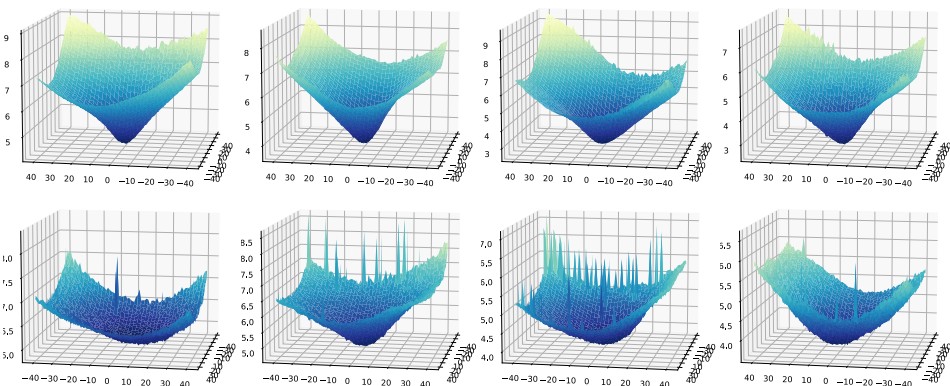

Figure 8: **Visualizing Loss Landscape (Uncurated).** Visualizing the loss landscape of StyleGAN inversion spanned by two principle directions. **Top** row shows 4 examples of the loss landscapes corresponding to our space $Z$. **Bottom** row shows the loss landscapes corresponding to the original input space $X$ for the same 4 examples. Note that the top row and the bottom row have different minimum loss values on the landscape because the minimas are obtained by performing independent optimization runs in space $Z$ and $X$ respectively. Given a fixed number of optimization step, optimization in $Z$ reaches lower loss values than optimization in $X$.

## B  EQUIVALENCE TO EXPECTATION-MAXIMIZATION

We show that our algorithm in 1 is equivalent to hard Expectation-Maximization (McAllester). First we rewrite the objective Eq. 2 as:

$$\hat{\theta} = \operatorname*{argmax}_{\theta} \max_{z_i} \prod_i P_\theta(y_i | F(\Theta(\mathbf{z}_i))) \tag{8}$$

To optimize objective Eq. 8, we perform the following algorithm:

1. Initialize $\theta$ with Gaussian distribution
2. Repeat the following until $\prod_i \ln P_\theta(y_i | F(\Theta(\mathbf{z}_i)))$ converges:
   (a) *Expectation*: $\rho_\theta(z_i) = \delta(z_i = \tilde{z}_i)$, where $\tilde{z}_i = \operatorname{argmax}_{z_i} P_\theta(y_i | F(\Theta(\mathbf{z}_i)))$
   (b) *Maximization*: $\hat{\theta} = \operatorname{argmax}_\theta \mathbb{E}_{z \sim \rho}[\ln(P_\theta(y_i | F(\Theta(\mathbf{z}_i))))]$

where $\theta$ denotes the parameters of the mapping network $\Theta$, and $\delta(\cdot)$ denotes a Dirac delta distribution. Since both expectation and maximization steps strictly increase or maintain the value of $P_\theta(y_i | F(\Theta(\mathbf{z}_i)))$ under gradient ascent, and the function is bounded, the algorithm is guaranteed to converge.

## C  DIMENSION OF $Z$

As our method learns a mapping network $\Theta$ that maps a vector from a new latent space $Z$ to a vector in the original input space $X$. The dimension of $Z$ becomes a hyperparameter. In table 3, we perform ablation studies on the effect of the dimension of $Z$ space on the performance of the learned mapping network. From table 3, we can see a consistent improvement on performance as we increase the number of dimension of $Z$. However, such improvement hits a diminishing return when the dimension is in the same order of magnitudes as the dimension of $X$.

| Dimension of $Z$ | 16 | 32 | 64 | 128 | 256 | 512 | 2048 | Baseline |
|---|---|---|---|---|---|---|---|---|
| In-distribution | 3.157 | 2.797 | 2.489 | 2.489 | 2.212 | 1.964 | 1.920 | 2.569 |
| OOD | 4.872 | 4.592 | 4.277 | 3.916 | 3.618 | 3.498 | 3.392 | 4.617 |

Table 3: Ablation studies on the dimension of $Z$. We trained variations of our full model with different number of dimension of $Z$ space, varying from 16 to 2048 given the dimension of input space $X$ is 512. Numbers correspond to loss values defined in Eq. 5.

## D  MORE EVALUATION ON OUT-OF-DISTRIBUTION GENERALIZATION

Since our proposed mapping network $\Theta$ is parameterized by a neural network, there's no guarantee that the learned mapping function is surjective. Therefore, we empirically study the generalization performance by testing a mapping network trained on CelebA-HQ against a spectrum of datasets from very similar ones (in-distribution) to completely different ones (OOD).

### D.1  SYNTHETIC SPECTRUM

We first created a synthetic version of this spectrum of datasets by varying the level of Gaussian noise injected into the images of CelebA-HQ. With a higher level of Gaussian noise injected into the original images, more of the original image content is corrupted, creating a distribution of images further away from the original test images.

| level of Corruption | $\mathcal{N}(0, 0.0^2)$ | $\mathcal{N}(0, 0.1^2)$ | $\mathcal{N}(0, 0.2^2)$ | $\mathcal{N}(0, 0.3^2)$ | $\mathcal{N}(0, 0.4^2)$ |
|---|---|---|---|---|---|
| Improvement | 29.53% | 18.95% | 12.57% | 6.30% | 6.72% |

Table 4: Evaluation of a mapping network trained with Celeba-HQ trainset and tested on the Celeba-HQ testset corrupted with different levels of Gaussian noise. The improvement is calculated by the percentage improvement of loss defined by Eq. 5 from baseline (no mapping network) to ours (with mapping network) after 200 steps of optimization.

### D.2  NATURAL IMAGE SPECTRUM

We then created a natural image version of this spectrum of datasets containing: CelebA-HQ (original in-distribution testset), AFHQ-Cat (center-aligned cat faces), LSUN-Cat (unaligned cat images), Container-Ship ("Container Ship" class from ImageNet), and Gaussian noise with individual pixel sampled from $\mathcal{N}(0.5, 0.5^2)$. From left to right, images in the datasets vary from very similar to human faces to very different.

From results evaluated on both synthetic spectrum and real spectrum, we observed consistent improvements of our method over the baseline, which shows the generalization performance of our method when applied to OOD data. From table 4 and 5, we see the improvements drop as the evaluation data is less and less in-distribution with the training data, which indicates that the mapping network does learn a prior from the training data.

| dataset | CelebA-HQ | AFHQ-Cat | LSUN-Cat | Container-Ship | $\mathcal{N}(0.5, 0.5^2)$ |
|---|---|---|---|---|---|
| Improvement | 29.53% | 35.99% | 24.36% | 16.87% | 6.60% |

Table 5: Evaluation of a mapping network trained with Celeba-HQ trainset and tested on a spectrum of datasets from the original CelebA-HQ (in-distribution) to Gaussian noise (OOD). AFHQ-Cat Choi et al. (2020) is a dataset with aligned cat faces. LSUN-Cat Yu et al. (2015) is a dataset with unaligned cat images. Container-Ship is 200 images sampled from "Container Ship" class of ImageNet Deng et al. (2009). The improvement is calculated the same way as table 4

## E    DEPENDENCE ON TRAINING DATASET

From previous section, we know that the learned mapping network contains priors learned from the training data. To exclude such influence, we train our mapping network using randomly generated images made of Gaussian noise and evaluate on testset of CelebA-HQ. From table 6, we see that even trained with the task of reconstructing images of Gaussian noise, our mapping network still bring some improvement over the baseline model, though much less significant. Interestingly, this improvement number is consistent with the last columnes of table 4 and 5, where a mapping network is trained with CelebA-HQ and tested on images with Gaussian noise.

| | Baseline | Ours (trained on Gaussian noise) | Improvement |
|---|---|---|---|
| CelebA-HQ | 2.569 | 2.396 | 6.72% |
| LSUN-Cat | 4.617 | 3.717 | 19.49% |

Table 6: Evaluation of a mapping network trained with images of Gaussian noise sampled from $\mathcal{N}(0.5, 0.5^2)$ clipped to [0, 1]. Numbers show the average loss defined in Eq. 5 evaluated on testset of CelebA-HQ and LSUN-Cat.

## F    LIMITATIONS

Optimization-based inference has intrinsic advantages to robustness, accuracy, and flexibility, which comes at the cost of additional computation time during inference. Encoder-based methods will usually perform faster because they only require a single forward pass of a neural network, while our approach requires several computational passes in both the forward and backward (gradient) direction. We believe that for many applications this trade-off will be desirable, especially in cases where accuracy is more important than speed. Our approach aims to minimize this additional computational overhead brought by optimization-based inference, and our experiments on multiple datasets show the significant computational savings compared to other optimization-based inference methods.

Unlike many other optimization-based inference algorithms, our approach also requires a training step in order to fit a suitable landscape, which requires both training time and training data. However, we believe this overhead is insignificant for most applications and we have designed our neural networks to be efficient. For example, $\Theta$ is relatively lightweight, making its training time fairly marginal compared to the training of the forward model $F$. In all our experiments, we found that the training time of $\Theta$ is orders of magnitudes faster than the training time for $F$.

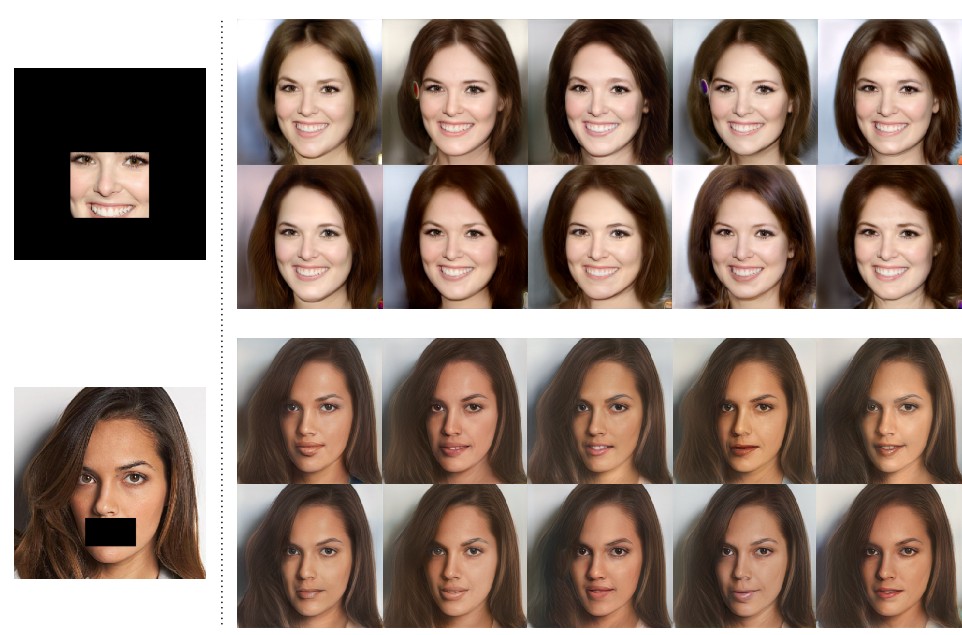

Partial GT     Reconstruction from Random Initializations

Figure 9: **Diversity of Masked Reconstructions.(Uncurated)** We visualize reconstructions for partially observable inputs from random initialization. The masked regions are not considered for loss computation, i.e., the gradient is set to be zero. By optimizing only on the partial observation, we obtain diverse, feasible solutions for the hidden regions. All reconstruction results presented are randomly sampled.

