# OpenReview forum: "Landscape Learning for Neural Network Inversion"
_ICLR.cc/2023/Conference — Submitted to ICLR 2023_

### Official Review · Reviewer_PZA2 · 2022-10-23

**Confidence:** 4
**Correctness:** 3
**Technical Novelty And Significance:** 3
**Empirical Novelty And Significance:** 4
**Recommendation:** 6

**Clarity, Quality, Novelty And Reproducibility:**

**Clarity:**
The paper is relatively well-written and has enough technical details. The illustrations are helpful and the presentation of results is clear.

**Quality:**
The technical part is simple and clear enough. It would be a plus if they can provide some theoretical explanation about why the proposed technique can learn better loss landscape (but I don't think the authors need to work on that during the rebuttal). The experiments include a wide range of problems and the results are interesting. However, I am not fully convinced because of the reasons stated in the "weaknesses" part.

**Novelty:**
Although the high-level idea of learning better latent space for optimization has also appeared in other subareas (e.g. [Meta-Learning with Latent Embedding Optimization](https://arxiv.org/abs/1807.05960)), applying this idea to neural network inversion is new as far as I know. The problems they are trying to solve are important because they are ubiquitous in applications.

**Reproducibility:**
The authors promised in the main text that they will release their code and model. No code has been provided yet.

**Details Of Ethics Concerns:**

I have no ethical concerns.

**Strength And Weaknesses:**

**Strength:**
1. The proposed technique is very simple yet it has been shown to be effective (greatly speeding up the inversion process) on several tasks involving neural network inversion.
2. The writing is easy to follow. The illustrations are helpful for understanding. The mathematical and experimental details are clear to me.

**Weaknesses:**
More empirical studies need to be done in order to support their claim and fully understand the proposed technique:
1. Firstly, does the learned latent space only depend on the model architecture, not the data? To what extent can we generalize out of domain? Can it be used to accelerate inference using the same model but with a *very* different dataset? I understand that the authors have already conducted experiments in the OOD scenario, but it would convince me even more if we can see the whole spectrum going from very similar datasets to completely different datasets (This analysis can be done using even synthetic data in a controlled setting). In addition, I would even suggest using random inputs to generate data and learn the latent space and run inferences using real data. It would be very interesting if we could still observe any improvement at all.
2. Secondly, it would be better if we could understand how much the learned latent space depends on the model architecture. For the purpose of analysis, what if we use the same dataset and a different model architecture?

I am not fully convinced by their statements regarding:
1. They mentioned that OBI supports generating diverse hypotheses while EBI can only generate one outcome. I think this might not be true as you can always inject noise into the encoder to generate diverse samples. One can also parameterize their encoders to output distributions over inputs.
2. They stated in the second paragraph that OBI can adapt without additional training. Will that still be true with the proposed learned new latent space or will the performance drop (at least slightly) with partially missing observations or new data distributions?

**Summary Of The Paper:**

This paper explores optimization-based neural network inversion. Instead of optimizing in the original input space, they propose to learn a new latent space and run optimization in the learned space during test time. They show that their technique greatly accelerates the inversion optimization process for a variety of tasks including GAN inversion, adversarial defense, and human pose reconstruction.

**Summary Of The Review:**

I recommend weak acceptance of this paper. On the one hand, this paper proposes a very simple technique that solves an important problem and they have applied their approach to a sufficient number of problems. On the other hand, I don't think I fully understand how their approach works and I think there should be more study (either theoretical justification or empirical study as suggested in the "weakness" section).

---

> ### Author Response · Authors · 2022-11-17
> **Response to Reviewer PZA2**
>
>
> We thank the reviewer for reading our paper carefully, raising important questions, and proposing very interesting experiments. We have conducted these experiments and have included them in the appendix of the paper with changes marked in blue. We respond below to your questions and concerns:
>
> > Q1: To what extent can we generalize out of domain? It would be
> > interesting to see the algorithm tested on a whole spectrum of
> > datasets from very similar ones to completely different ones.
>
> This is a great suggestion! We have included section D in the appendix to study this quantitatively. Specifically, we have created both synthetic and real datasets which form spectrums from datasets that are most similar to the training data, and datasets that are completely different. From both experiments, we see a drop in improvements from most similar to completely different test data. However, the improvements of our method over the baseline are consistent, indicating our method is able to generalize out-of-distribution. More importantly, the generalization performance of optimization-based methods (i.e., OBI) is superior to that of encoder-based methods as shown in Figure 3 and 7, and this paper focuses on improving the optimization efficiency of OBI.
>
> > Q2: Proposed experiments: train the model with randomly generated
> > noise for reconstruction, then test on inference dataset
>
> This is also a great suggestion! We have included section E in the appendix to study this quantitatively. We do see an improvement, though much less significant. Interestingly we found that the model trained with Gaussian noise and tested on CelebA-HQ sees a similar improvement as the model trained with CelebA-HQ and tested on Gaussian noise.
>
> > Q3: Proposed experiments: compare different model architectures on the
> > same dataset.
>
> Thank you for the question. It is ambiguous which “model” the reviewer is referring to. If the reviewer is referring to the forward model, we have applied the mapping network to 3 different architectures on different tasks, including both generative and discriminative models. This indicates that our method is general across different forward model architectures. If the reviewer is referring to the architecture of the mapping network, we have added quantitative studies in section C in the appendix to compare the performance of mapping networks with different architectures. From these results, we observe that the performance of the learned latent space is dependent on the architecture choice of the mapping network. Poor architecture choice, for example, setting the dimension of space $Z$ to be significantly lower than that of the input space $X$ will lead to worse performance than the baseline method. We leave more experiments regarding the architecture design of the mapping network to future work.
>
> > Q4: the statement that EBI can only generate one outcome is not true –
> > variational encoders that output distribution
>
> Thank you for pointing this out. We agree and have removed the last sentence of section 2.1 where the original statement is located.
>
> > Q5: Will the performance of the proposed method drop for
> > out-of-distribution data or partially missing observations?
>
> Thank you for the question. As stated above, we have conducted extensive experiments evaluating our method’s generalization performance over a spectrum of datasets, from which we observe consistent improvements of our method over the baseline. In terms of data with partially missing observations, please see Figures 6 and 9 for results of image inpainting and outpainting as well as masked reconstruction of 3D humans. Our method produces diverse and high-fidelity results from partially missing observations.

---

> > ### Comment · Reviewer_PZA2 · 2022-11-23
> > **Response to Authors**
> >
> > Thank you for conducting the additional experiments. These results look very interesting. Even when they learn the landscape on random noise, the proposed technique can still bring performance gain during test time. It suggests that at least part of the performance gain is data-independent. By "comparing different model architectures on the same dataset", I actually mean, if you learn the landscape together with forward model 1, can the learned landscape be directly used with forward model 2 on the same dataset? Sorry for the confusion. Because my response comes after the rebuttal deadline, this should not be seen as the authors' failure to address my concern. In general, the authors have done a great job running analysis, and I am much more convinced that their proposed techniques are useful for our community. I would recommend acceptance but I am afraid I will keep my current score for now.

---

> > > ### Author Response · Authors · 2022-11-27
> > > **Further Discussions**
> > >
> > > We are glad that our response addressed your concerns and that our updated paper made you much more convinced about the contribution of our work. Thank you for clarifying the question regarding the model architecture. While this is a very interesting question, we think it is slightly outside of the scope of the paper. It is possible that the proposed mapping network could learn from data-dependent and architecture-independent patterns to help improve optimization across different architectures out of the box, but we are not claiming this without further experimental results. Given the variability of architectures of neural networks and different assumptions and loss functions used to train them (e.g. with or without Gaussian prior, regularization etc.), it is hard to conduct scientific experiments accordingly. In addition, the dimensions of the input space between two neural networks are often different. For example, BigGAN's latent space dimension is 128 and StyleGAN's is 512, making them incomparable. We thank the reviewer again for their engagement in the discussion and helpful feedback.

---

### Official Review · Reviewer_A9dv · 2022-10-25

**Confidence:** 4
**Correctness:** 3
**Technical Novelty And Significance:** 2
**Empirical Novelty And Significance:** 3
**Recommendation:** 5

**Clarity, Quality, Novelty And Reproducibility:**


Concerned about incremental novelty.

Most of the paper appears to be reproducible, although unless someone must experiment with the
code, else it may be difficult to completely judge.


**Strength And Weaknesses:**


Paper is easy to follow and read. Enough of empirical results provided to substantiate the claim.


Algorithm 1 is not referred in text; although trivial matter - bit it should be done at least once.
"MAML" is not specified anywhere in document- pl, expand the term.

Is your method similar to applying ADMM over a manifold/topological space, to optimize ?

Would have preferred more analytics concerning the convergence of your algorithm  - does that have any minimal guarantee ?

You talk of patterns of trajectories being learned from latent space.
 How many such pattern clusters exist? DO they sufficiently represent categories of populations required
with no bias?


Few references not cited, some example:
Landscape and training regimes in deep learning; Mario Geiger, Leonardo Petrini, Matthieu Wyart; Physics Reports, Volume 924, 15 August 2021, Pages 1-18.

Neural Network Inversion in Adversarial Setting via Background Knowledge Alignment; Ziqi Yang et al; Proceedings CCS '19; Neural Network Inversion in Adversarial Setting via Background Knowledge Alignment; https://dl.acm.org/doi/abs/10.1145/3319535.3354261

T. Hospedales, A. Antoniou, P. Micaelli and A. Storkey, "Meta-Learning in Neural Networks: A Survey" in IEEE Transactions on Pattern Analysis & Machine Intelligence, vol. 44, no. 09, pp. 5149-5169, 2022.
doi: 10.1109/TPAMI.2021.3079209

TENGRAD: TIME-EFFICIENT NATURAL GRADIENT DESCENT WITH EXACT FISHER-BLOCK INVERSION; Saeed Soori et. al;
https://www.cs.toronto.edu/~mmehride/papers_private/TENGRAD.pdf






**Summary Of The Paper:**

The paper introduces a method that learns a loss landscape
where gradient descent is efficient, bringing massive improvement and acceleration
to the inversion process of learning a CNN. Advantages are demonstrated using a number of methods
for both generative and discriminative tasks.

The method helps to accelerate optimization-based inference to invert a forward model. The concept involves
learning a new space that is easier than the original input space to optimize with gradient descent at testing time.


**Summary Of The Review:**



Inversion of FF models will help solving many inverse problems.
The proposed method may help researchers to explore more work in this direction.

The lack of sufficient analytics is a major cause of worry.

---

> ### Author Response · Authors · 2022-11-17
> **Response to Reviewer A9dv**
>
>
> Thank you for the careful reading and helpful feedback. We respond below to your questions and concerns:
>
> > Q1: is the method similar to ADMM over a manifold/topological space?
>
> Thank you for the question. Indeed, both ADMM and our method are alternating optimization algorithms. However, our optimization is not subject to linear equality constraints, and the objectives for the alternating update of $\theta$ and $z$ are not decoupled, which are two defining characteristics of ADMM.
>
> > Q2: Does the algorithm have a convergence guarantee?
>
> Thank you for the question. Following your comment, we have included section B in the appendix to show the equivalence between our algorithm and hard Expectation-Maximization and provided convergence analysis. Note that our convergence guarantee is for local minima as a global minima guarantee is extremely difficult for optimizations of neural networks under moderate assumptions. Empirically, we have observed consistent convergence to results that outperform the baseline across 3 tasks with tens of experiments conducted across the paper.
>
> > Q3: How many pattern clusters of optimization trajectories exist? Do
> > they cover the population without bias?
>
> Thank you for the question. Performing clustering analysis in the loss landscape over the input space of a neural network remains an open research problem [1][2][3][4] that we will leave for future work. Regarding population coverage and bias, we’d like to acknowledge that the mapping network in our method is trained with publicly released datasets. The forward models we used are also publicly available pretrained models. These datasets and models likely contain biases that our method inherits. We believe this is an important matter, and we have discussed it in more detail in our ethics statement (section 6).
>
> > Q4: Most of the paper appears to be reproducible, although unless
> > someone must experiment with the code, else it may be difficult to
> > completely judge.
>
> We have provided all necessary implementation details (see section 7) to reproduce the method presented in our paper. In addition, we will release all code, models, and data to make sure the results presented in the paper are easily reproducible.
>
> > Comment 1: Algorithm 1 is not referred to; MAML is not specified.
>
> Thank you for pointing these out. Both issues are addressed in the updated paper. MAML stands for Model-Agonistic Meta-Learning. We have specified this in the paper.
>
> > Comment 2: References not cited.
>
> Thank you for the reminders. We have included these in our related work section of the updated paper.
> \
> \
> [1] Smith, Leslie N., and Nicholay Topin. "Exploring loss function topology with cyclical learning rates." arXiv preprint arXiv:1702.04283 (2017).
>
> [2] Goodfellow, Ian J., Oriol Vinyals, and Andrew M. Saxe. "Qualitatively characterizing neural network optimization problems." arXiv preprint arXiv:1412.6544 (2014).
>
> [3] Li, Hao, et al. "Visualizing the loss landscape of neural nets." Advances in neural information processing systems 31 (2018).
>
> [4] Im, Daniel Jiwoong, Michael Tao, and Kristin Branson. "An empirical analysis of deep network loss surfaces." (2016).

---

### Official Review · Reviewer_4P2g · 2022-10-25

**Confidence:** 4
**Correctness:** 3
**Technical Novelty And Significance:** 2
**Empirical Novelty And Significance:** Not applicable
**Recommendation:** 6

**Clarity, Quality, Novelty And Reproducibility:**

### Clarity
The paper is clearly written and easy to follow in general.
### Novelty
The proposed method is marginally novel, using a neural network as a surrogate of latent space.
### Reproducibility
Codes are not provided, but sufficient implementation details are given.

**Strength And Weaknesses:**

### Strength
* The paper is well-written and easy to follow.
* The motivation is clear, and the intuition behind the proposed method is straightforward.
* The experiment results seem promising. Inverting a neural network has a lot of advantages and downstream applications. The proposed * method appears effective and can further accelerate and stabilize such a process.
### Weaknesses
My primary concern is regarding the sub-optimal solution when using the proposed method. Is there any circumstance of Z that may result in a suboptimally averaged solution since the loss landscape is a smoothed-out version of X? If so, it is reasonable, as it can be considered a tradeoff between speed and quality. However, showcasing such examples and quantitatively measuring the tradeoffs would be better.

**Summary Of The Paper:**

The paper proposes a framework to accelerate and stabilize the process of neural network inversion. Specifically, the proposed method uses a neural network to learn a mapping between the latent spaces that allows gradient descent over the input to converge in fewer steps. Experiments show that the proposed framework can successfully improve the convergence speed for optimization.

**Summary Of The Review:**

I think this is an interesting paper to accelerate neural network inversion. Although the proposed method is rather simple, it seems to be effective. I am willing to raise my recommendation if the question mentioned in the weakness section is addressed.

---

> ### Author Response · Authors · 2022-11-17
> **Response to Reviewer 4P2g**
>
> Thank you for writing a thoughtful review and recognizing many strengths of our paper while raising some concerns. Please let us know if the following response has sufficiently addressed these concerns. We are happy to engage in any further discussions or provide clarifications.
>
> > Q1: Since the intuition behind the mapping network is smoothing of the
> > loss landscape, are there cases where optimization may result in a
> > suboptimally averaged solution?
>
> Thank you for the question. From empirical studies, we have not observed such suboptimally averaged solutions. In addition to the quantitative results shown in Figure 3, which shows the average loss across all test data, we went on to check the statistics of the actual percentage of test data among which our reconstruction outperforms that of the baseline:
>
> | GAN-inversion |  | 3D Human | |
> |--|--|--|--|
> |in-distribution|100.0%|in-distribution|100.0%
> |OOD| 99.5%|OOD| 87%
>
> In other words, in nearly every case, our method is better than the baseline. Additionally, results in Figure 4 (step 2000) and 5 show that our reconstructed images and human pose are not suboptimally averaged solutions. They contain high-frequency details such as earrings, wrinkles, and the texture of hair which are faithful reconstructions of the observations.
>
> We want to also mention that there is indeed a tradeoff between training and inference time associated with our method. Our method requires the training of an extra mapping network, which accelerates the optimization-based inference significantly.

---

> > ### Comment · Reviewer_4P2g · 2022-12-04
> > **Response to Authors**
> >
> > Thank you for the response. Although there is no theoretical justification, I think the empirical studies provided are sufficient enough to address my concern. Therefore, I would recommend acceptance.

---

### Official Review · Reviewer_AFoq · 2022-10-25

**Confidence:** 3
**Correctness:** 4
**Technical Novelty And Significance:** 3
**Empirical Novelty And Significance:** 3
**Recommendation:** 6

**Clarity, Quality, Novelty And Reproducibility:**

**Clarity:** The paper was well-written and easy to follow.

**Novelty:** To the reviewer's knowledge, incorporating gradient descent trajectories in the training of an encoder for neural network inversion is novel.

**Reproducibility:** No code is provided, but training and implementation details are provided.

**General comments:**
- In regards to one of my earlier comments about the applicability of the method, I had a question about the following scenario. One useful thing about typical GAN priors are that they are agnostic to the forward model (given measurements $y = f(x) + \eta$, one could use a GAN $G$ to reconstruct $x$ by solving $\min_z ||f(G(z)) - f(x)||$ by gradient descent). Thus, in principle this GAN could be used for different forward models $f(\cdot)$ and solve different inverse problems (over the same image distribution). The proposed method, however, learns an encoder so that optimization over the input of $f\circ G$ is easier. This also requires examples of outputs $y$. Could one, instead, train the encoder to be GAN specific and then use it for multiple forward models down the line? For example, could one train the encoder to invert a StyleGAN with gradient descent and then be used at inference time to solve an inpainting or deblurring problem?
- How does the size of the input space of the encoder play a role in the results? For example, does having a lower latent dimension in the encoder aid/hurt the results (potentially by having less representational capacity)?


**Strength And Weaknesses:**

**Strengths:**
- The proposed training strategy allows for gradient descent to be performed more quickly in the latent space in that reasonable solutions are found with fewer iterations.
- The new encoder performs better on out of distribution images for inversion than standard encoder-based inference approaches.

**Weaknesses:**
- The way the methodology is proposed, the encoder would be specific for the forward model considered, rather than simply to the GAN or generator being inverted. I wonder if the authors could comment on a scenario that could help broaden the methods scope (in the next section).


**Summary Of The Paper:**

This work considers the problem of neural network inversion, which lies at the core of many downstream tasks incorporating neural networks, such as inverse problems. Naively inverting a network via gradient descent is challenging, due to the induced non-convex landscape with potentially many local minima. The authors present a method that takes a pretrained neural network $F$ and learns an encoder $\Theta$ such that gradient descent is easier to perform over the input space of $F \circ \Theta$, as opposed to directly over the input space of $F$. The model is trained by minimizing a loss over many trajectories of gradient descent for a fixed number of steps. The authors apply their approach to GAN inversion, 3D pose reconstruction, and protecting against adversarial attacks. The proposed approach shows that gradient descent empirically requires fewer steps to approach a solution as compared to direct inversion.

**Summary Of The Review:**

Overall, I found the ideas in the paper intuitive and the results look fairly compelling.

---

> ### Author Response · Authors · 2022-11-17
> **Response to Reviewer AFoq**
>
> We thank the reviewer for reading our paper in detail, raising important questions, and proposing very interesting experiments. We have conducted these experiments and have included them in the appendix of the paper with changes marked in blue. We respond below to your questions and concerns:
>
> > Q1: Proposed mapping network is specific to the type of task that was
> > used for training. What about training a mapping network on one task
> > (reconstruction), and applied to a different task such as inpainting
> > or deblurring problems?
>
> Thank you for the question. We have conducted experiments where the mapping network is trained for reconstruction and applied to both inpainting and outpainting tasks. Please see Figures 6 and 9 for the results. Additionally, we have conducted more quantitative experiments on the generalization performance of the mapping network in the appendix. We have shown that when tested on a spectrum of datasets varying from very similar to completely different ones, our method consistently outperforms the baseline. Please see sections D and E in the appendix for these results.
>
> > Q2: How does the size of the input space of the mapping network play a
> > role in the results?
>
> This is a great question! We got curious about this question as well, which led us to conduct an ablation study on the dimension of space $Z$ (see section C in the appendix). We tested 7 ablated variations of our model with input dimensions varied from 16 to 2048, given 512 is the dimension of space $X$. We see consistent improvements in performance as the dimension of input space increases, but with a diminishing return when the number of dimensions is in the same order of magnitude as the input space $X$.

---

### Author Response · Authors · 2022-11-17
**General Response to All Reviewers**

We thank the reviewers for taking the time to read and review the paper. We are glad the reviewers found our paper to be novel (R1), widely applicable (R2), well-written (R3), and interesting (R4). We’d like to reiterate that the problem we are studying – neural network inversion is a ubiquitous problem in deep learning research including many applications in computer vision, robotics, graphics, etc. In this paper, we propose a practical approach to accelerate the process of optimization-based neural network inversion and demonstrate the massive improvement in optimization efficiency brought by our method. In addition, we have added several experiments suggested by the reviewers which help improve the paper. We believe our contribution will bring a significant impact on relevant research directions.

---

### Author Response · Authors · 2022-12-06
**Sincerely looking forward to further discussions**

We thank all the reviewers' effort in reading and evaluating this work. We particularly appreciate reviewer 4P2g and PZA2 for acknowledging our rebuttal and engaging in further discussions. We understand the other reviewers might have been busy with other engagements and the NeurIPS conference, but we still very much look forward to any constructive feedback for our work and rebuttal.

We would like to highlight the additional analysis (Appendix B) and experiments (Appendix C, D, E) we've added to the paper. Specifically, in section B of the appendix, we've provided a theoretical analysis showing the equivalence of our alternating optimization algorithm to hard EM, and provided convergence analysis, as requested by reviewer A9dv. In section C of the appendix, we've conducted ablation studies to understand the effect of dimension of the learned space $Z$ on the final performance as requested by reviewer AFoq. In section D and E, we've conducted several experiments to study our method's dependence on dataset priors and generalization performance as requested by reviewer PZA2. These analysis and experiments have improved the soundness and completeness of our work and we appreciate the reviewers' helpful suggestions for that.

Despite the best of efforts, confusion may still exist with only a single round of interaction. With these new experimental results, we warmly welcome any additional questions or discussions to address any remaining concerns or potential misunderstandings. We believe that the reviewers, the AC and us can reach a consensus on our work at the end, and we sincerely hope to hear from you soon.

---

### Decision · Program_Chairs · 2023-01-20

**Decision:**

Reject

**Justification For Why Not Higher Score:**

See above comments. Reviewers mildly positive, this case was discussed with the meta AC and we decided to reject the paper for now.

**Justification For Why Not Lower Score:**

N/A

**Metareview: Summary, Strengths And Weaknesses:**

This work considers the problem of neural network inversion, which is an important problem in the field of machine learning (for instance, it is quite popular for inverting GANs). The authors claim that naive inversion with gradient descent might be prone to having difficulties due to highly non-convex landscapes. They present a method that takes a pretrained neural network and learns an encoder such that gradient descent is easier to perform over the transformed space defined by the encoder.

The reviewers were mildly positive about the paper (6,6,6,5) and did not change their opinion after the discussion period. This paper was therefore discussed with the meta-AC and we concluded this paper is not yet ready for acceptance.

Although the paper presents some encouraging empirical results, there are several important problems that remain:
- The paper does not provide any strong evidence explaining why the proposed mapping yields a more efficient method. It is for instance not clear whether Figure 2 is accurate in any way. The only justification is a set of PCA visualizations of the landscape in Appendix A, but this is very preliminary and incomplete evidence.

- The paper ignores a large body of literature that enforces smoothing constraints on the latent space of probabilistic models, see for instance the seminal paper by Tipping and Bishop:
Tipping, Michael E., and Christopher M. Bishop. "Probabilistic principal component analysis." Journal of the Royal Statistical Society: Series B (Statistical Methodology) 61.3 (1999): 611-622.
http://www.cs.columbia.edu/~blei/seminar/2020-representation/readings/TippingBishop1999.pdf

- This is more minor but there is a lot of work in the field of optimization that shows that SGD can optimize complex non-convex landscapes, I would also expect a deeper discussion in the paper.

In summary, I'm afraid I'm not able to recommend acceptance at this stage. I strongly encourage the authors to address these two main points before resubmitting their paper to another venue.

**Summary Of Ac-Reviewer Meeting:**

See the above answer:
1) The paper ignores a large body of literature.
2) The paper does not provide sufficient evidence to support its claims.